# In situ reversible underwater superwetting transition by electrochemical atomic alternation

Qianbin Wang[1], Bojie Xu[1], Qing Hao[2], Dong Wang[2], Huan Liu[1] & Lei Jiang[1,3]

Materials with in situ reversible wettability have attractive properties but remain a challenge to use since the inverse process of liquid spreading is normally energetically unfavorable. Here, we propose a general electrochemical strategy that enables the in situ reversible superwetting transition between underwater superoleophilicity and superoleophobicity by constructing a binary textured surface. Taking the copper/tin system as an example, the surface energy of the copper electrode can be lowered significantly by electrodeposited tin, and be brought back to the initial high-energy state as a result of dissolving tin by removing the potential. Tin atoms with the water depletion layer inhibit the formation of a hydrogen-bonding network, causing oil droplets to spread over the surface, while copper atoms, with a high affinity for hydroxyl groups, facilitate replacing the oil layer with the aqueous electrolyte. The concept is applicable to other systems, such as copper/lead, copper/antimony, gold/tin, gold/lead and gold/antimony, for both polar and nonpolar oils, representing a potentially useful class of switchable surfaces.

[1] Key Laboratory of Bio-Inspired Smart Interfacial Science and Technology of Ministry of Education, School of Chemistry, Beijing Advanced Innovation Center for Biomedical Engineering, Beihang University, 100191 Beijing, P. R. China. [2] Key Laboratory of Molecular Nanostructure and Nanotechnology, Institute of Chemistry, Chinese Academy of Sciences, 100190 Beijing, P. R. China. [3] CAS Key Laboratory of Bio-inspired Materials and Interfacial Science, Technical Institute of Physics and Chemistry, Chinese Academy of Sciences, 100190 Beijing, P. R. China. These authors contributed equally: Qianbin Wang, Bojie Xu. Correspondence and requests for materials should be addressed to H.L. (email: liuh@buaa.edu.cn)

Superwettable surfaces provide effective solutions to many academic and industrial problems[1–7], e.g. controllability in printing or patterning[8,9]; robustness in antifogging, anti-fouling, or self-cleaning materials[10,11]; and environmental protection in the petroleum industry[12,13]. Although many attempts have recently been made to trigger the wettability transition by applying light illumination[14,15], temperature[16], solvents[17], electrical potential[18], or pH stimulus[19], the wettability transition between superphilicity and superphobicity usually requires an offline treatment, causing most of these approaches to be ex situ. Other technologies that allow an in situ wettability alternation, such as electrowetting[20] and electrochemical reduction/oxidation[21], only work within a limited scale with a contact angle (CA) variation from ~50° to ~110° or from <10° to ~90° and remain unable to enjoy the aforementioned superwetting advantages. To date, in situ reversible superwetting transitions (i.e., CA changes between ~0° and ~180°) have been far from effective. Here, by using an electrochemical approach, we successfully realize the in situ reversible superwetting transition between a perfectly spherical-shaped droplet and a completely spread liquid film state by switching off and on a tiny potential.

## Results

**In situ reversible underwater superwetting transition**. A typical in situ reversible underwater superoleophilic–superoleophobic transition is schematically and experimentally presented in Fig. 1 (Supplementary Movie 1). Here a textured copper (Cu) substrate, with typical microscale clusters and nanoscale papillae, was used; the substrate was prepared by electrochemical deposition (Fig. 1a, b, Supplementary Figure 1). When a liquid droplet of 1,2-dichloroethaneoil (~2 μL) was released onto the textured Cu electrode in an aqueous electrolyte (consisting of 0.5 M $H_2SO_4$ and 0.01 M $SnSO_4$, Methods) without applying a voltage, it presented a perfectly spherical shape with a CA of ~180° (underwater superoleophobicity; Fig. 1d, 0 s). Upon applying a voltage of −0.5 V, the oil droplet started to spread on the electrode surface and realized complete spreading within <1 s, with the CA quickly declining from ~180° (underwater superoleophobic state) and approaching 0° (underwater superoleophilic state) (Fig. 1d). Of note is that the fully spread oil thin film can in situ turn back to the original spherical shape by removing the voltage (Fig. 1e), whereby the in situ conversion from superphilicity to superphobicity is first realized experimentally. As shown, when the applied potential was off, the Cu electrode presented a light-gray area beneath the oil film (red arrows in Fig. 1e, 0 s), and then the spread oil film suddenly retracted, along with the expansion of the light-gray area (Fig. 1e, 32.8 s). The oil film continuously retracted with time and finally presented a spherical shape with a CA of ~180° within 115 s (Fig. 1e). The recovered oil droplet could even roll away from the electrode surface (Fig. 1f), showing low adhesion to the substrate. Quantitatively, the droplet's height ($H$) (as indicated in Fig. 1g) presented a quick decay from 1.5 to ~0.2 mm by applying a potential, and then automatically rose to 1.475 after ~170 s of removed voltage (Fig. 1c). Specifically, the transition from fully spread oil film to spherical droplet is a spontaneous process without any external energy or forces. Thus we here provide an electrochemical approach to realize the in situ reversible superwetting transition between superphilicity and superphobicity (Fig. 1g).

**Electrochemical characterization**. We next explored the electrochemical process involved in this in situ reversible superwetting transition by cyclic voltammetry (CV) and electrochemical quartz crystal microbalance (EQCM) measurements (Fig. 2a, b). The underpotential deposition (UPD) of tin

(Sn)[22,23], $Sn^{2+} + 2e^- \rightarrow Sn_{UPD}$, started at a nucleation potential of ~−0.33 V (vs. Ag/AgCl) (Fig. 2a, peak a in the red curve). As the potential became more negative, the current density increased sharply past −0.46 V (Fig. 2a, peak b in the red curve), which is assignable to the bulk deposition (BD) of Sn[22–24], $Sn^{2+} + 2e^- \rightarrow Sn_{bulk}$. The EQCM results suggested a two-stage mass increase on the electrode at these two potential regions (Fig. 2b, black curve, points a and b), as clearly visualized by the differentiated curve (Fig. 2b, blue curve, points a' and b')[22]. Here the hydrogen evolution reaction (HER), $2H^+ + 2e^- \rightarrow H_2$, started at ~−0.65 V (Fig. 2a, the red curve), which is more negative than that on the pure Cu electrode by ~ 0.25 V (Fig. 2a, the black curve)[22]. We found that the superwetting transition could only be realized in a certain potential range negative to −0.46 V (Fig. 2a, the blue curve in the gray region), where the BD of Sn took place in the aqueous electrolyte. We confirmed that the surface wettability remained unchanged when only the UPD of Sn occurred in the potential region positive to −0.46 V for the 0.5 M $H_2SO_4$ + 0.01 M $SnSO_4$ electrolyte or no Sn electrodeposition occurred in the potential region negative to −0.46 V for the 0.5 M $H_2SO_4$ electrolyte (Supplementary Figure 2). In addition, no wettability change was observed when Sn was electrodeposited in a non-aqueous ionic liquid electrolyte (Supplementary Figure 3). We therefore propose that the main parameters governing the in situ reversible underwater superwetting transition are the aqueous electrolyte and the electrodeposited Sn layer (Supplementary Figure 4), which could be further dissolved into the solution as $Sn^{2+}$ by switching off the potential (Supplementary Figure 5, 6). Here the reversible deposition and dissolving of atomic Sn on the Cu electrode was directly confirmed by the EQCM characterization. As shown, the mass on the electrode presented a clear and sharp increase as a result of the Sn deposition and then gradually decreased to the original value because of the Sn dissolving into the solution (Supplementary Figure 6). Here the residual of Sn was approaching 0 after numerous cycles under the experimental conditions, indicating the good reversibility of the Sn deposition/dissolving. Consequently, the in situ superwetting transition enjoys a rather good reversibility with no observable decay in the CA (Supplementary Figure 7a). The deposition of Sn is a typical interfacial electrochemical process, which happens very fast, while the dissolving of Sn is a diffusion-controlled process that is normally slow and heavily dependent on the local environment[25]. Thus a strategy that can accelerate the diffusion of $Sn^{2+}$ is advantageous for the conversion from philic to phobic states. Specifically, the conversion time from philic to phobic states could be clearly shortened by increasing the temperature, a result of the enhanced $Sn^{2+}$ diffusion (Supplementary Figure 7b).

**In situ Raman spectra**. By in situ Raman spectroscopy, we obtained direct evidence that the electrodeposited Sn layer was capable of altering the hydrogen-bonding network at the electrolyte/electrode interface. To visualize the Raman peaks associated with hydrogen bonding, a $D_2O$ electrolyte with 0.5 M $H_2SO_4$ + 0.01 M $SnSO_4$ was specifically used. As shown in Fig. 2c, three peaks are assignable to the hydrogen groups on the surface, where two are from the chemical bonds of O-D and D-O-D on the surface (Fig. 2c, peaks A–C)[26] and one is related to the $SO_4^{2-}$ (Fig. 2c, peak D)[27]. Clearly, the peak intensities of these four peaks were drastically enhanced when the potential was off (Fig. 2c, curve 2) and largely weakened when the potential was on (Fig. 2c, the curve 1). This phenomenon clearly suggests that the density and orientation of the water molecules on the electrode surface were drastically changed via application of a potential, as schematically shown in Fig. 2d. Here, during the electrochemical

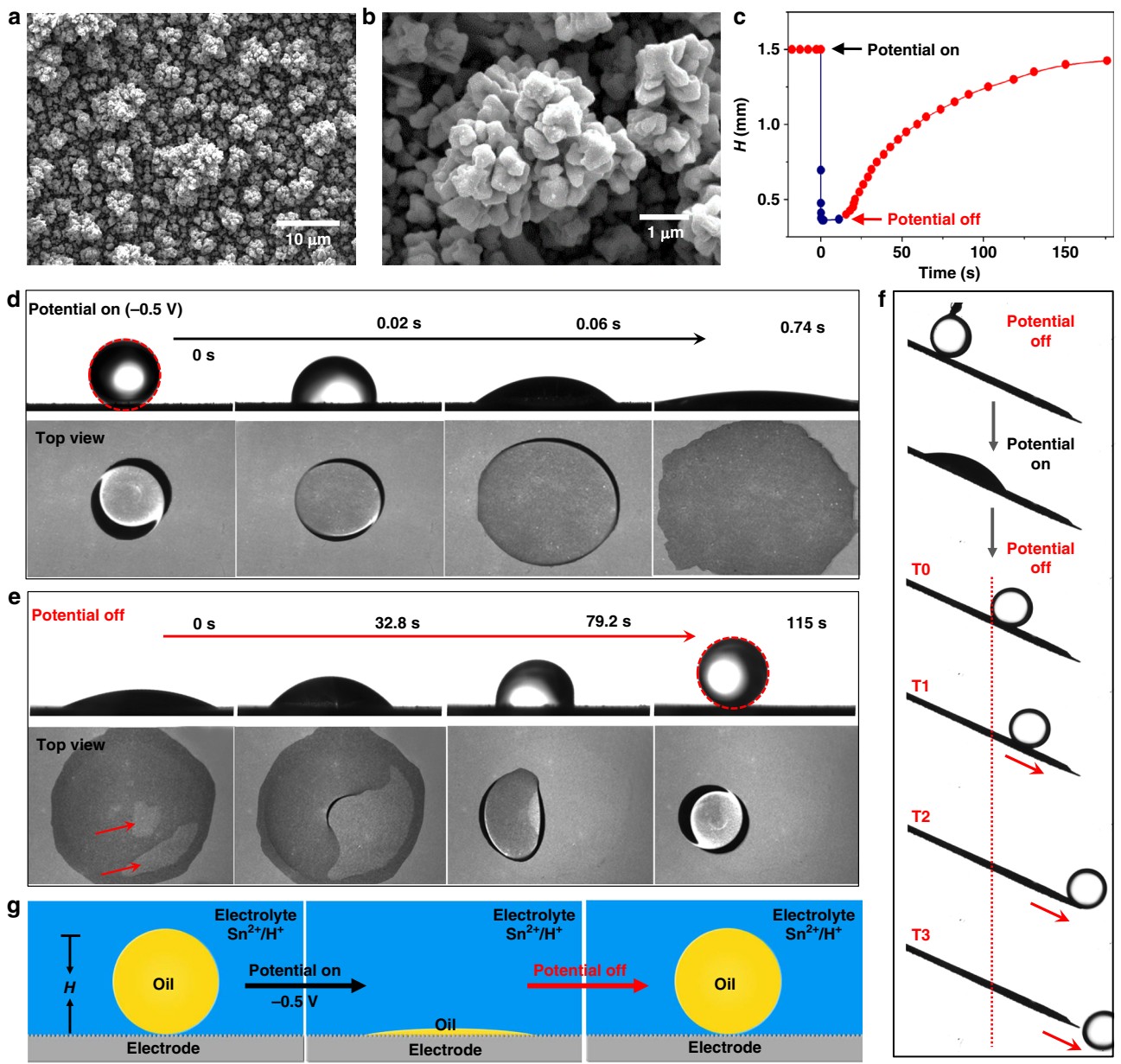

**Fig. 1** In situ reversible superwetting transition between underwater superoleophilicity and superoleophobicity. **a**, **b** Scanning electron microscopic images of the copper electrode show the typical hierarchical micro-/nanostructures. **c** The corresponding heights of the oil droplet during the underwater superoleophobic–superoleophilic–superoleophobic transition in the presence and absence of the voltage. **d**, **e** In situ characterization of the morphology variation for a 2 μL 1,2-dichloroethaneoil droplet on a rough copper electrode. The side and top views **d** for the oil spreading process when the potential is on (i.e., the transition from underwater superoleophobicity to superoleophilicity) and **e** for the process where the spread oil retracts into a spherical shape when the potential is off. **f** The oil drop can even roll away from a tilted substrate after an in situ reversible wettability conversion of superphobicity–superphilicity–superphobicity. **g** A drawing illustrating the whole in situ reversible superwetting conversion

deposition process, the enrichment of Sn atoms on the rough Cu surface (Supplementary Figure 5) inhibited the formation of the hydrogen-bonding network at the electrode/water interface (Fig. 2d, the left column) since Sn atoms on the electrode exhibited a lower tendency to exchange electrons and form hydrogen bonds with interfacial water molecules[28–30]. The as-deposited Sn atoms with the water depletion layer enabled the formation of hydrophobic domains[1,28–30], turning the electrode into a superhydrophobic surface. Thus the oil experienced a higher tendency to spread on the Sn-enriched electrode surface than on the aqueous electrolyte, exhibiting underwater super-oleophilicity. After removing the potential and as a result of dissolving Sn into the electrolyte as $Sn^{2+}$ (Fig. 2d, the right column), the electrode gradually recovered to the original Cu-enriched surface (Supplementary Figure 5), showing a high affinity for the hydroxyl group[31]. Therefore, the aqueous electrolyte spreading on the Cu electrode becomes preferred, leading the surface to return to the superoleophobic state. This metal–metal $(M_1/M_2)$ binary composite surface, composed of a high-surface-energy underlayer of Cu $(M_1)$ and a low-surface-energy deposited layer of Sn $(M_2)$[32], enabled a drastic alteration of the underwater surface hydrogen-bonding network depending on the amount of each atom, making the surface wettability reversibly changeable in situ. More importantly, we confirmed that such a binary composite metal surface is valid with other systems such as Cu/Pb, Cu/Sb, Au/Sn, Au/Pb, and Au/Sb—where Pb, Sb, and Au

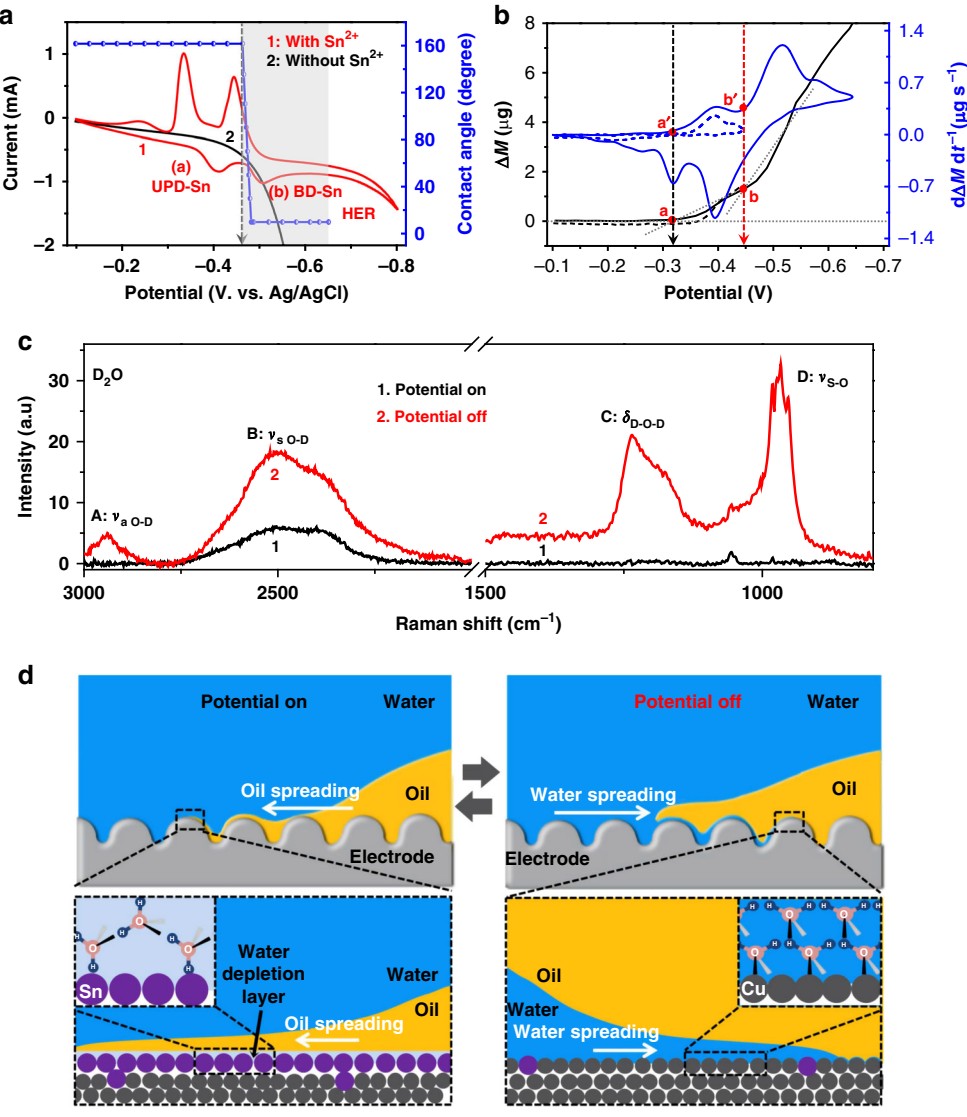

**Fig. 2** The binary composite surface governs the in situ reversible superwetting transition. **a** Cyclic voltammetry of a Cu electrode in $H_2SO_4/SnSO_4$ (red line) and $H_2SO_4$ electrolyte (black line). Scan rate: 20 mV s$^{-1}$. **b** The electrochemical quartz crystal microbalance evaluation of the gravimetric alteration during this in situ reversible superwetting transition. **c** The in situ Raman spectrum characterization of hydrogen bonding on the electrode surface when the potential was switched on and off. Peaks A and B are assignable to the intermolecular symmetric and antisymmetric stretching vibrations of the hydrogen bond O-D, respectively; peak C is the bending vibration of the D-O-D bond; and peak D is related to the symmetric stretching vibration of the S-O bond. **d** The schematics show that the Cu/Sn binary composite surface enables drastic altering of the hydrogen-bonding network at the electrolyte/electrode interface, making the oil droplet experience the in situ reversible transition between superphobicity and superphilicity. When the potential is on, the enrichment of Sn atoms inhibits the hydrogen-bonding network on the electrode surface, turning the electrode into the underwater superoleophilic state; when the potential is off, the Cu presents a high affinity to the hydroxyl group, leading the surface to return to its intrinsic superphobic state

represent lead, antimony, and gold, respectively (Supplementary Figure 8, 9)—and thus is representative of a general approach for realizing in situ superwetting transitions.

**Theoretical calculation of energy**. We compared the interfacial energies of systems that were wetted by either oil ($E_o$) or water ($E_w$) to understand the switchable wettability on the electrode (Supplemental notes 1, 2 and Supplementary Figure 10a). Based on the minimization of the underwater system's free energy[33], the energetic gap, $\Delta E = E_w - E_o$, was a decisive factor in determining whether the electrode preferentially interacted with the water or the oil under certain conditions. From the calculated result (Supplementary Note 1), we have $\Delta E > 0$ when applying a voltage, which indicates that the rough electrode will be

wetted preferentially by oil when the potential is on, while $\Delta E < 0$ is obtained after removing the potential, suggesting that the oil can be substituted by water to form a more stable system when the potential is off.

The system's total free energy governs this superphobic–superphilic transition (Supplementary Figure 10b), which can be minimized into, $E_{sys} = (A - A_0 \cos\theta_{OS})\gamma_{OE}$, where the Cu/Sn composite surface is regarded as an integrated solid (S) with a fluctuating wettability, and $A_0$ and $A$ are the areas of the O/S interface and the O/E interface, respectively. When the oil droplet is deposited on the electrode surface in a oleophilic state, we can consider that it makes a spherical cap of radius $R_1$, in which the volume of the oil droplet is $\Omega_1 = \frac{2}{3}\pi R_1^3[1 - \frac{3}{2}\cos\theta_{OS} + \frac{(\cos\theta_{OS})^3}{2}]$, and the total surface energy

can be expressed as $E_1 = \pi R_1^2 (1 - \cos\theta_{OS})^2 (2 + \cos\theta_{OS})\gamma_{OE}$. In the superoleophobic state, for the spherical oil droplet with radius $R_0$, the volume is $\Omega_0 = 4\pi R_0^3/3$, and the total surface energy of the oil droplet is $E_0 = 4\pi R_0^2 \gamma_{OE}$. The condition $\Omega_0 = \Omega_1$ implies $R_0 = [\frac{1}{2} - \frac{3}{4}\cos\theta_{OS} + \frac{(\cos\theta_{OS})^3}{4}]^{1/3} R_1$. Hence, the surface energy index ($E_1/E_0$) can be written as:

$$E_1/E_0 = 0.63[(1 - \cos\theta_{OS})^2(2 + \cos\theta_{OS})]^{1/3} \qquad (1)$$

The surface energy index curve shows a descending trend with the CA decreasing (Fig. 3a, the solid black curve), suggesting that tuning the surface energy by an electrodeposited Sn layer can spontaneously induce a wettability transition from superphobic to superphilic states (CA changes from ~180° to ~0°, from state 1 to 1' in Fig. 3a). Here the CA transition from a superphilic state (~0°, state 2 in Fig. 3a) to a superphobic state (~180°, state 2' in Fig. 3a) is remarkable since the inversion of the spontaneous superphobicity–superphilicity transition process is energetically

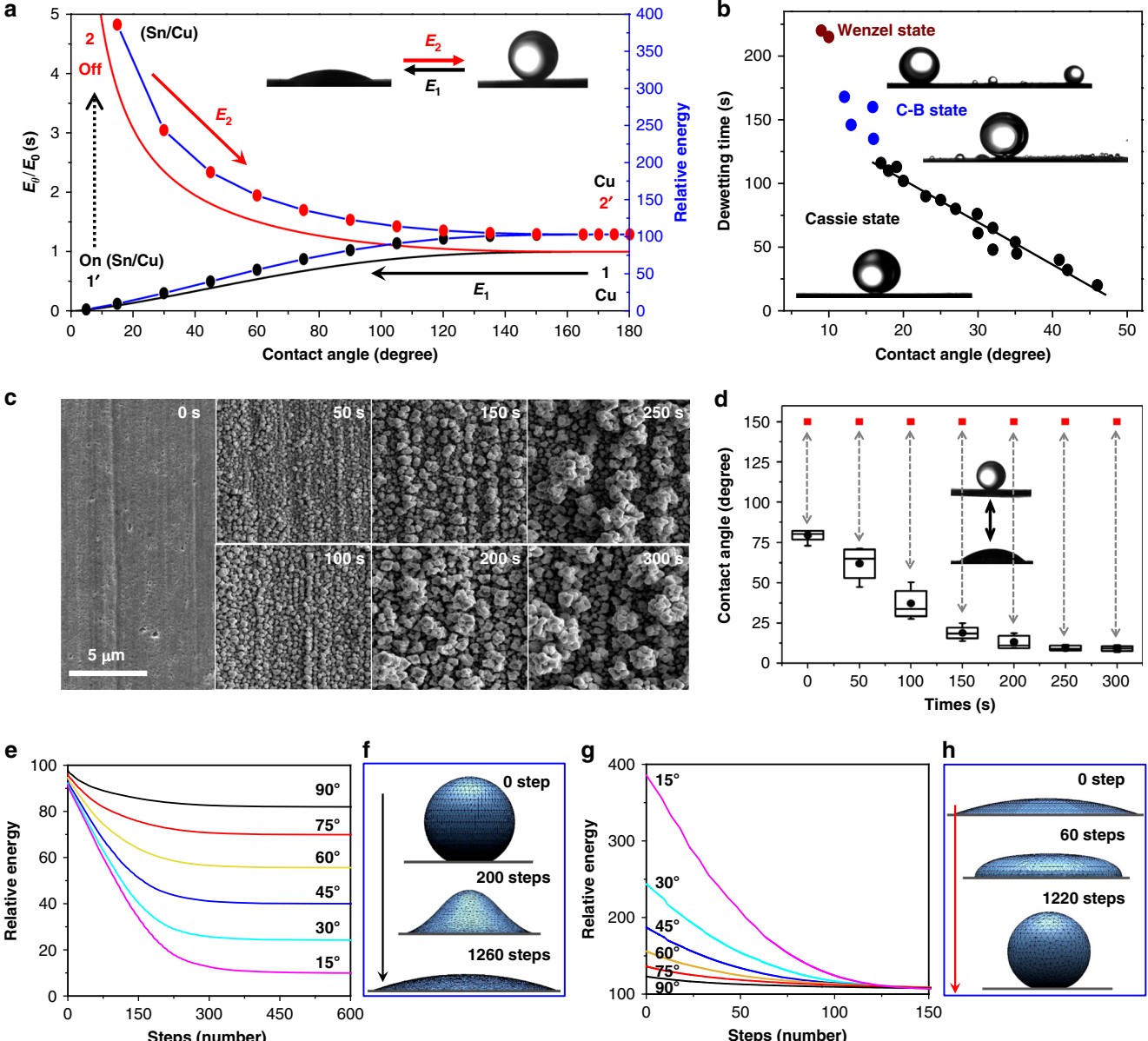

**Fig. 3** The theoretical calculation and the wetting transition within different ranges. **a** The surface energy index of a droplet with various contact angles (CAs) by the theoretical calculation (solid curves) and by the hybrid-energy-minimization (HEM) calculation (dotted curves). $E_0$ is the initial surface energy of a droplet with a CA of 180°, and $E_1$ and $E_2$ are the surface energies of droplets with CA transitions from ~180° to ~0° and from ~0° to ~180°, respectively. **b** Three typical wetting states involved in the in situ reversible superwetting transition: Cassie wetting, Cassie–Baxter wetting, and Wenzel wetting. **c** Scanning electron microscopic images of Cu electrodes with different roughnesses obtained by varying the electrochemical deposition time of Cu from 0, 50, 100, 150, 200, 250 to 300 s. **d** The wetting transition can be varied between the superoleophobic (CA > 150°) state and a wide range of oleophilic (CA from ~78° to ~0°) states on the surface with different roughnesses. The data are presented as box-and-whisker plots; box, 25th–75th percentile; whiskers, 5th and 95th percentile expansion; solid black circle, mean. **e, f** Variations in the relative total surface energy of a 5-pL droplet with an initial CA of 150° released on a homogeneous surface with different wettability values (with CAs of 15°, 30°, 45°, 60°, 75°, and 90°) as a function of the number of steps for the HEM calculation. **g, h** Variation in the relative total surface energy of a 5-pL droplet placed on a homogeneous surface with a CA of 150° as a function of the number of steps for the HEM calculation. The initial CAs are 15°, 30°, 45°, 60°, 75°, and 90°, respectively

unfavorable[34]. However, in our electrochemical system, the presence and absence of an electrodeposited Sn layer empowers the conversion of the substrate's wettability, making this in situ reversible superwetting transition possible. By removing the voltage, the Sn layer on the electrode immediately starts to dissolve into the solution (from state 1' to 2 in Fig. 3a), and with time, the surface gradually recovers to its initial superoleophobic state (from state 2 to 2' in Fig. 3a). Here the surface energy index ($E_2/E_0$) is:

$$E_2/E_0 = \frac{2 - 2\cos\theta_{OS} + (\sin\theta_{OS})^2}{0.63[(1-\cos\theta_{OS})^2(2+\cos\theta_{OS})]^{2/3}} \qquad (2)$$

The surface energy index curve of this condition suggests the energetically spontaneous process of the wettability transition from a complete spreading state to a perfect spherical shape (Fig. 3a, the solid red curve).

We then utilized a hybrid-energy-minimization (HEM) technique to evaluate the energy of oil with different apparent CAs on the solid surface[35,36]. The result presents a similar trend to that of the surface energy index, which provides convincing evidence to support the theoretical calculation (Fig. 3a, the dotted black and red lines). To simulate the in situ superphobic–superphilic transition, the complete spreading of a 5-pL oil droplet on a homogeneous substrate was monitored; the apparent CA changed from a value of ~180° at the initial contact state to ~0° at the equilibrium state. During the HEM calculations, the total energy decreased exponentially as the steps increased, and finally, the droplet wetted the substrate with an equilibrium CA of ~0° (when the relative energy residual was estimated at <10$^{-6}$) (Fig. 3a, the black dotted line). Here, by switching off the potential, the surface chemical nature of our system transformed immediately (i.e., from state 1' to state 2 in Fig. 3a) as a result of the instant change in the surface components. Thus the total energy variation could be monitored by taking the change in the CA on the substrate from ~0° to 180°. Specifically, the relative total energy increased suddenly to >400. Then the total energy gradually decreased with increasing step number, during which the dewetting of a fully spread oil film into a spherical drop was associated with an equilibrium CA of ~180° (Fig. 3a, the red dotted line). The droplet's total energy after the energy minimization calculation agrees well with the theoretical calculation, suggesting that this superphobic–superphilic transition is an energetically favorable process[37].

**Effect of surface roughness on in situ reversible underwater wettability transition.** In general, wettability on a certain surface can vary depending on the surface roughness[38]. Here we demonstrated that the in situ transition range of surface wettability could be varied between superoleophobicity (CA > 150°) and a wide range of oleophilic states (CA from ~78° to ~0°) by altering the surface roughness (Fig. 3c, d). With prolonged electrodeposition times of Cu including 0, 50, 100, 150, 200, 250, and 300 s (Fig. 3c), the steady oil CA obtained after applying a voltage (CA$_{+vol}$) showed a rapid decline of ~78.9°, 61.9°, 43.4°, 21.9°, 15.4°, 11.0°, and 10.5°. Of note is that the oil CA on all these surfaces could recover to the initial superphobic state (Fig. 3d, red points) after removing the potential. The HEM calculations also support the results (Supplementary Note 2). For the transition from superoleophobicity to oleophilicity, the total energy presented the same exponentially decreasing trend when the substrates with various inherent CAs from 15° to 90° (Fig. 3e, f) were used. When the substrate became superoleophobic, the total energy decreased with increasing step number for droplets with

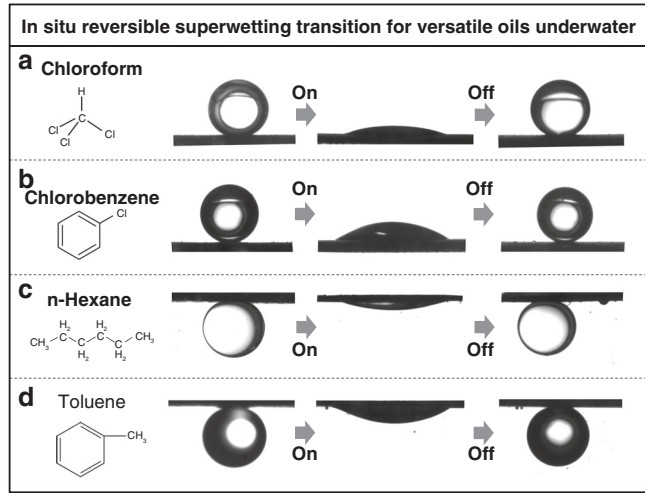

**Fig. 4** Generality of the in situ reversible underwater superwetting transition for various oils. Various oils including polar (e.g., toluene, chlorobenzene, and chloroform) and nonpolar (e.g., n-hexane) oils were observed to participate in the in situ reversible superwetting transition, which demonstrates that this in situ reversible superwetting transition can be shared by a range of oils underwater

different initial CAs, and ultimately, all droplets reached the same stable superphobic state (Fig. 3g, h).

Further investigations on the dewetting time of oil droplets with various CA$_{+vol}$ revealed three typical states: the Cassie state, Cassie–Baxter state (C-B state), and Wenzel state (Fig. 3b)[39]. For the Cassie state, where the spread oil slightly penetrate into the microtexture, only a single large droplet was observed to sit on the electrode surface after recovery, and the dewetting time linearly decreased with increasing CA$_{+vol}$. When the oil film was partly pinned at the microtexture, i.e., the C-B state, a relatively large droplet, surrounded by many small satellite droplets, was obtained because the oil film would split during the recovery process. If the oil was fully immersed in the highly rough site (the Wenzel state), numerous small droplets formed. For CA$_{+vol}$ < 20°, an apparent increase in dewetting time was observed, suggesting the oil film began to enter the microtexture (Fig. 3b). This result indicates that the substrate roughness can significantly affect the oil's final recovery state, providing a feasible approach for manipulating liquids in a controllable manner.

**Generality and application of manipulating droplets.** We also demonstrated that this in situ reversible superwetting transition could be shared by various oils, including polar (e.g., toluene, chlorobenzene and chloroform) and nonpolar (e.g., n-hexane). As shown in Fig. 4, the in situ reversible superwetting was clearly observed for oil droplets with densities both higher (e.g., chloroform and chlorobenzene) and lower (e.g., n-hexane and toluene) than that of water. This set-up therefore represents a class of switchable surfaces, which show versatile possibilities in manipulating liquids. For example, the substrates can be applied in oil collection/recovery by a typical oil adsorption–desorption strategy, which has, as an advantage, the regeneration of a fresh copper surface after manipulating the oil droplets. Not only the adsorption and spreading of oil droplets on the substrate but also the recovery and releasing of oil droplets from the substrate are enabled by the same substrate, which means that the substrate is free from being befouled by the adsorbed oil. Specifically, we demonstrated two typical behaviors by using copper fibers: a single oil droplet was adsorbed and fully spread on a copper mesh with an applied voltage, and then, the oil was recovered and

released as numerous tiny droplets by switching off the potential (Supplementary Figure 11a, Movie 2); numerous tiny oil droplets were sequentially adsorbed and spread on a copper fiber with an applied voltage and then recovered and released as a single droplet by switching off the potential (Supplementary Figure 11b, Movie 3). Behaving like a dropper pipette, the fibers with the in situ reversible superwetting transition can controllably collect and release droplets, providing an alternate strategy for manipulating droplets.

## Discussion

In summary, by constructing a binary Cu/Sn composite surface with a certain texture, we successfully realized the in situ reversible superwetting transition between a perfectly spherically shaped droplet (superphobicity) and a completely spread liquid film (superphilicity) upon the switching off and on of a potential. The electrodeposited Sn layer induces a water depletion layer by inhibiting the hydrogen-bonding network at the water/electrode interface, leading to the spreading of both polar and nonpolar oil droplets underwater. After removing the potential, the original Cu-enriched surface is gradually recovered as a result of dissolving Sn into the electrolyte as $Sn^{2+}$ and shows a high affinity to hydroxyls. The energy calculation demonstrates that all processes involved are energetically favorable. In particular, the as-prepared binary composite metal surface that enables the in situ reversible superwetting conversion is also applicable to other systems, such as Cu/Pb, Cu/Sb, Au/Sn, Au/ Pb, and Au/Sb. Here we propose a general electrochemical strategy for realizing the in situ reversible superwetting conversion, which might open an avenue for designing and fabricating various smart materials or devices for manipulating liquids in various applications, such as microfluidics; oil recovery in the petroleum industry; drop control in printing and patterning; and smart antifogging, antifouling, or self-cleaning materials.

## Methods

**Electrochemical experiments**. The electrochemical experiments were carried out by using a CHI-1240B electrochemical workstation in a homemade beaker-type cell. A $0.5 \times 2$ cm L-shaped rough Cu foil, achieved by electrochemically depositing Cu film on a polished Cu foil at $-1.0$ V in an aqueous electrolyte containing 0.5 M $CuSO_4$ and 0.15 M $H_2SO_4$ was used as the working electrode. A Pt plate was used as the counter electrode, and an Ag/AgCl electrode was the reference electrode. The aqueous solution of 0.5 M $H_2SO_4$ and 0.01 M $SnSO_4$ was used as the electrolyte. CV measurements were performed from $-0.1$ to $-0.8$ V with a scan rate of 10 mV s$^{-1}$. The in situ reversible superwetting transitions were carried out by releasing a 2-μL oil droplet (1,2-dichloroethaneoil, toluene, chlorobenzene, chloroform, or n-hexane) on the rough copper electrode. The shape alterations of the oil droplet on the electrode were recorded by CCD cameras (acA1600–60gm) taking images at 25 frames per second. The dissolving of Sn on the Cu electrode and the consequent diffusion of $Sn^{2+}$ were further modified by either increasing the temperature or enhancing the local mass transfer using a flowing system with stirring.

**Characterizations of the surface morphology and elements of the electrode**. The surface morphology of the Cu electrode with and without the electrochemically deposited Sn layer was evaluated by an environmental scanning electron microscope (SEM, Quanta 250 FEG) and a JEOL JSM-7500F SEM. The mapping of surface elements was recorded by an energy-dispersive X-ray spectroscopy analyzer. X-ray photoelectron spectroscopy (ESCAlab 220i-XL) was used to analyze the chemical elements on the surface of the electrode.

**In situ electrochemical Raman spectra**. Raman spectra were collected with a Microscope Raman Spectroscopy (Thermo Scientific DXR) excited by a laser of 633 nm; light water was replaced with $D_2O$ (Aladdin, D >99.9%) in this measurement. The electrochemical measurements, using a CHI-1240B electrochemical workstation, were carried out in an electrochemical cell with a rough Cu foil as the working electrode. The in situ Raman spectra of the surface of the working electrode were collected while switching on/off the potential.

**Electrochemical quartz crystal microbalance**. The EQCM measurements were performed using an EQCM-D (Q-Sense E4) system equipped with an EQCM module (QEM 401). The quartz crystal, AT-cut, was coated with a 300-nm-thick

layer of copper metal (Q-Sense). The electrochemical measurements were carried out in an electrochemical cell using a three-electrode arrangement, with the QCM crystal as the working electrode, a Pt plate as the counter electrode, and Ag/AgCl as the reference electrode. The shift in the quartz resonance frequency ($\Delta f$) was converted into the mass change ($\Delta m$) on the copper-coated quartz during cycling by applying Sauerbrey's equation:

$$\Delta m = -\frac{\sqrt{\rho_q \mu_q}}{2f_0} \cdot \Delta f = -C_f \cdot \Delta f \tag{3}$$

where $\rho_q$ is the density of quartz, $\mu_q$ is the shear modulus of quartz, $f_o$ is the fundamental resonance frequency of the quartz, and $C_f$ is the calibration constant (or sensitivity factor).

## Data availability

The authors declare that the data supporting the findings of this study are available within the paper and its Supplementary Information or from the corresponding author upon reasonable request.

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

## Acknowledgements

We are grateful for the financial support from the National Natural Science Foundation of China (21622302, 21872002, 21574005) and the Fundamental Research Funds for the Central Universities. We appreciate the helpful discussion with Professor Shuji Nakanishi in Osaka University.

## Author contributions

H.L. and L.J. conceived and supervised the project. H.L., B.X. and Q.W. designed the experiments. B.X. and Q.W. contributed equally to this work. B.X., Q.H. and D.W. performed the in situ electrochemical Raman spectra experiments. Q.W. and B.X. wrote the manuscript. H.L. and L.J. revised the manuscript. All authors discussed the results and commented on the manuscript.

## Additional information

**Competing interests:** The authors declare no competing interests.

**Journal Peer Review Information**: *Nature Communications* thanks the anonymous reviewers for their contribution to the peer review of this work

