## [Peer Review File · Nature Communications]

Reviewers' comments:

Reviewer #1 (Remarks to the Author):

This work by the group of Lei Jiang represents a remarkable achievement. Even though there are plenty of papers showing tunability in wetting, the overwhelming majority of them requires the surface to be dry during the trigger. The small number of papers dealing with tunable wetting while the droplet remains in contact with the surface only allow relatively small changes in contact angle. Here the authors show a huge change in contact angle, that is fast and reversible. They used an electrochemical reaction to deposit a thin layer of tin that changes the interaction between oil droplet and surface. This demonstration is an enormous achievement, and will encourage the search for fast and huge contact angle changes in other wetting systems. Furthermore, the experimental proof is nearly complete. I have only a few comments that I would like to see addressed. Otherwise I would recommend its publication in Nature Communications.

1. Figure S6 shows that mass increases rapidly upon switching potential on, and mass loss when potential is off, corresponding to the formation and disintegration of the Sn layer. In a single cycle of switching on and off, the mass is however not recovering to the original value. For every new on-off cycle the mass at the end of the cycle increases seemingly linearly as function of cycle. Why? What happens? Is there still some amount of tin remaining after potential is off? If not all tin is removed after switching potential off, does it mean that the reversibility is limited to only a few cycles? I would like to see how contact angle changes for a large number of cycles, plotted as contact angle vs number of cycles.
2. Figure 1 shows that the speed of a phobic to philic transition is very fast (less than a second), whereas it takes more than 100 seconds to do the reverse transition. Could the reverse transition be speeded up?
3. Figure S2, rolling off of droplet is important and in my view this figure could move to the main text.

Reviewer #2 (Remarks to the Author):

Comments on "In Situ Reversible Underwater Superwetting Transition by Electrochemical Atomic Alternation"

The manuscript describe an electro-deposition/ dissolution system for superoleophilic and superoleophobic property transition. In general, the manuscript is solid. However, as an application oriented research project, the content can be more attractive if some application prospective is presented. Detailed comments are listed:

Comment 1: The characterization is solid. In order to make the system more interesting, a potential application utilizing the system should be presented. Clarify why transition from 0 to 180 degree is more beneficial than those reported ones.

Comment 2: The free energy analysis is not solid. In the supporting information, the free energy change (dG) should be expressed as: $dG=R(\gamma_{ws}-\gamma_{os})dS_{os}-\gamma_{ol}dS_{ol}$. dS_{os} is different from dS_{ol} . The energy analysis part should be rewritten. Moreover, energy expression in Page 5 is not correct. For example, " $E_0=4\pi R^2$ " should be " $E_0=Y*4\pi R^2$ ".

Point-by-point responses to the specific comments of the reviewer:

Responses to the reviewer #1

Comment 1: Figure S6 shows that mass increases rapidly upon switching potential on, and mass loss when potential is off, corresponding to the formation and disintegration of the Sn layer. In a single cycle of switching on and off, the mass is however not recovering to the original value. For every new on-off cycle the mass at the end of the cycle increases seemingly linearly as function of cycle. Why? What happens? Is there still some amount of tin remaining after potential is off? If not all tin is removed after switching potential off, does it mean that the reversibility is limited to only a few cycles? I would like to see how contact angle changes for a large number of cycles, plotted as contact angle vs number of cycles.

Answer 1: Thanks for the helpful suggestions. According to the reviewer's comment, the reversibility of the *in situ* superwetting conversion was carefully examined, and the data was shown in Figure S6 and Figure S7. We confirmed that the deposition and dissolving of atomic Sn on the Cu electrode was reversible for multiple times by EQCM characterization (Supplementary Fig. S6), and consequently numerous cycles of the *in situ* reversible superwetting transition is shown (Supplementary Fig. S7a) without observable decay of contact angle. **Here, the remaining amount of Sn after each cycle was directly related to the duration time of potential off, as well as the diffusion process of the Sn²⁺ in the solution, and it can be significantly reduced by increasing the duration time of potential off (Figure S6, Figure R1).** Specifically, when the duration time of potential off was prolonged from 180 s to 1400 s, the remaining amount of Sn was drastically decreased from 1 μg to about 0.2 μg, as shown in Figure R1. Given the enough long duration time of potential off and the ideal diffusion of Sn²⁺ from the Cu electrode surface, the residual of Sn can tend to 0, by which numerous cycles of such reversible superwetting conversion is possible. Experimentally, trace amount of Sn remaining after cycles of "deposition-dissolving" was reasonable under the operation condition, since the diffusion of Sn²⁺ from the electrode is a rather dynamic and complicated process. However, such trace amount of Sn has little effect on the superwetting

conversion here. Particularly, the remaining amount of Sn was only 0.04% after 5 cycles operation (Figure S6b). As far as the coverage of Sn on the Cu electrode surface lower than 0.41%, the surface always show superoleophobicity (Figure R1c). Hence, in theory, the *in situ* superwetting transition is reversible for at least 50 cycles. In case that the Sn^{2+} can be remove off the electrode immediately, more cycles reversibility is reasonable.

Figure R1. (a, b) Gravimetric measurements of Sn electrodeposited on a Cu electrode with different duration time of potential off. It clearly showed that the remaining amount of Sn can be significantly reduced by increasing the duration time of potential off. (c) EDX mapping of Sn element on the Cu electrode. The electrode was superoleophobicity when the percentage of residual Sn on the electrode was 0.41%.

We have carefully revised the manuscript, see marked paragraph in page 5 and 6.

Comment 2: Figure 1 shows that the speed of a phobic to philic transition is very fast (less than a second), whereas it takes more than 100 seconds to do the reverse transition. Could the reverse transition be speeded up?

Answer 2: Thanks for the helpful comment. Yes, as the reviewer point out, the speed of a phobic to philic transition is very fast, because it is triggered by a typical interfacial electrochemical process. However, the conversion from a philic to a phobic state is a diffusion controlling process, which is normally slow and heavily depending on the local environmental (*J. Am. Chem. Soc.* 75, 555-559, 1953). In addition, because the oil droplet spreading on the atomic Sn layer can significant hamper the diffusion process of Sn^{2+} because of the physical spacial confine. Therefore, strategies that can enhance the diffusion process is advantage to

speed up the conversion from a philic to a phobic state. Here, we demonstrate that increasing the temperature can drastically fasten the conversion from a philic to a phobic state, as summarized in Figure S7b. Specifically, when the $\Delta\theta$ was 140° , increasing the temperature from 25°C to 80°C , the conversion time was largely shortened from about 185 s to 110 s. We have carefully revised the manuscript, see marked paragraph in page 5 and 6.

Comment 3: Figure S2, rolling off droplet is important and in my view this figure could move to the main text.

Answer 3 : Thanks for the high evaluation, and we have moved the information in Figure S2 to the main text Figure 1f.

Responses to the reviewer 2

Comment 1: The characterization is solid. In order to make the system more interesting, a potential application utilizing the system should be presented. Clarify why transition from 0 to 180 degree is more beneficial than those reported ones.

Answer 1: Thanks for the suggestions. Here, based on the *in situ* superwetting transition, a typical “adsorption-desorption” strategy in oil collecting/recovery application was presented (Figure S9), which takes big advantages on the regenerating of the fresh copper surface after manipulating oil droplets. Not only the absorption and spreading of oil droplets on the substrate, but the recovery and releasing of oil droplets from the substrate were enables at the same substrate, which means the substrate is free from being befouled by the adsorbed oil. Particularly, we demonstrated two typical behaviors of controllable manipulating oil droplets by using copper fibers. (1) A single oil droplet was adsorbed and fully spread on a poised copper mesh, then was recovered and released as numerous tiny droplets by switching off the potential (Fig. S9a, Movie S2). Specifically, a $5\ \mu\text{L}$ oil droplet was adsorbed and totally spread on the copper mesh surface when a -0.5V voltage is applied. After removing the voltage, the completely adsorbed oil grandly retracted into numerous tiny droplets, and then drip off, finally,

be released/recovered from the copper mesh automatically. (2) Numerous tiny oil droplets were sequentially adsorbed and spread on a poised copper fiber, then were recovered and released as a single droplet by switching off the potential (Fig. S9b, Movie S3). Using a copper fiber with certain potential, numerous tiny oil droplets were absorbed and spread on the copper fiber in a controllable manner, forming a continuous oil film on the copper fiber (as indicated by dashed line in Figure S9-b6). After removing the potential, the continuous oil film retracted into a big droplet gradually and then released off from the fiber. Behaving like a dropper pipette, the fibers with the in situ reversible superwetting transition can controllably collect and release droplets, providing a new strategy for manipulating droplets.

We have carefully revised the manuscript, see marked paragraph in page 11.

Comment 2: The free energy analysis is not solid. In the supporting information, the free energy change (dG) should be expressed as: $dG=R(\gamma_{ws}-\gamma_{os})dS_{os}-\gamma_{ol}dS_{ol}$. dS_{os} is different from dS_{ol} . The energy analysis part should be rewritten. Moreover, energy expression in Page 5 is not correct. For example, “ $E_0=4\pi R^2$ ” should be “ $E_0=Y*4\pi R^2$ ”.

Answer 2: Thanks for the kind suggestion. We are very sorry for it and we have carefully revised the calculation of the free energy analysis. In the supporting information, the parameter of apparent contact area of the oil-solid interface S_{os} and contact area of the oil-water interface S_{ow} were introduced to the free energy change expression. And energy expression in line 7 and line 9 paragraph 1 page 8 have been revised.

Reviewers' comments:

Reviewer #1 (Remarks to the Author):

The manuscript has improved but not yet to a sufficient level.

In their response authors write "We confirmed that the deposition and dissolving of atomic Sn on the Cu electrode was reversible for multiple times by EQCM characterization (Supplementary Fig. S6), and consequently numerous cycles of the in situ reversible superwetting transition is shown (Supplementary Fig. S7a) without observable decay of contact angle."

Looking at the figure, "reversible for multiple times" means for the authors 3 times. And in fact as I had commented before, the figure shows the process is not fully reversible. It may be a small difference but the difference is real and could become significant for large number of cycles.

Additionally authors write they did "numerous cycles" of superwetting transitions "without observable decay" of contact angle. According to the authors numerous means 6 times. From the figure I would not claim there is no decay of contact angle, because the number of data points is too small. More data points would be needed for a statistically meaningful conclusion.

In their comment authors point out that "in theory the transition is reversible for at least 50 cycles". The theory authors use is an oversimplification. Instead manuscript would be greatly improved if the mass change and contact angle change is demonstrated experimentally for at least 50 cycles. Reversibility is a important concept in this work that can be seen from the title, and should be corroborated by a credible experimental demonstration.

In Fig S7a important experimental parameters are missing, including times of switching.

In my view Figure S6 should contain the ΔM data for the two different times of off state, similar as is shown in the Figure R1.

Authors added the sentence "The concept is applicable to other systems as Cu/Pb, Cu/Sb, Au/Sn, Au/ Pb and Au/Sb for both polar and nonpolar oils, representing a new class of switchable surfaces.". This is a relevant addition showing generality. The only added data to back up this claim is the table in Figure 2d3 and it shows only qualitatively that transition does occur. Do these systems reach the same contact angle values? Manuscript would be significantly improved if actual data of these systems is shown, such as contact angle vs cycles.

Supplementary movie 3 could show time indication.

Reviewer #2 (Remarks to the Author):

My previous comments have been well addressed. I have no further comments.

Point-by-point responses to the specific comments of the reviewer:

Responses to the reviewer #1

Comment 1: In their response authors write “We confirmed that the deposition and dissolving of atomic Sn on the Cu electrode was reversible for multiple times by EQCM characterization (Supplementary Fig. S6), and consequently numerous cycles of the *in situ* reversible superwetting transition is shown (Supplementary Fig. S7a) without observable decay of contact angle.” Looking at the figure, “reversible for multiple times” means for the authors 3 times. And in fact as I had commented before, the figure shows the process is not fully reversible. It may be a small difference but the difference is real and could become significant for large number of cycles.

Answer 1: Thanks for the comments, and we are very sorry for confusing the reviewer. Here, we perform the new experiment to carefully check the reversibility of such *in situ* superwetting transition, where the dissolving of Sn on the Cu electrode and the consequent diffusion of Sn²⁺ in surrounding was modified by either increasing the temperature or enhancing the local mass transfer using a flowing system with stir (i. e., increasing flowing rate of the electrolyte in EQCM experiment). As shown in Figure S6, we confirmed that the deposition and dissolving of atomic Sn on the Cu electrode was reversible for multiple times by EQCM characterization. Here, the deposition amount of Sn differs because the slightly different time of potential on in each cycle. As shown in the magnified inset picture in Figure S6, the mass on electrode show clear and sharp increase as a result of Sn deposition, which then gradually decrease to the original value because of the Sn dissolving. It indicates that the residual of Sn was approaching to 0 after numerous cycles under experimental condition, indicating a good reversibility of the Sn deposition/dissolving. Consequently, the *in situ* superwetting transition enjoys a rather good reversibility with no observable decay of the contact angle (Supplementary Fig. 7a).

We have carefully revised the paper, and please see paragraph in page 5.

Comment 2: Additionally authors write they did “numerous cycles” of superwetting transitions “without observable decay” of contact angle. According to the authors numerous means 6 times. From the figure I would not claim there is no decay of contact angle, because the number of data points is too small. More data points would be needed for a statistically meaningful conclusion. In their comment authors point out that “in theory the transition is reversible for at least 50 cycles”. The theory authors use is an oversimplification.

Answer 2: Thanks for the helpful comment. As described before (see Answer 1), we have performed new experiments to carefully check the reversibility of such *in situ* superwetting transition. As shown in Figure S7a, the superwetting transition is reversible for large number of cycles (over 50 cycles) with no clearly decay of CA under experimental conditions where the diffusing of Sn²⁺ was enhanced by using a flowing system with stir.

We have carefully revised the paper, and please see paragraph in page 5.

Comment 3: In Fig S7a important experimental parameters are missing, including times of switching. In my view Figure S6 should contain the delta M data for the two different times of off state, similar as is shown in the Figure R1.

Answer 3 : Thanks for the suggestions. The curve of contact angle vs. cycles was used in Figure S7a to show the reversibility of the *in situ* superwetting conversion. As reviewer pointed out, the time of switching is also an important parameter, which, however, have been clearly pointed out in Figure 1. In our system, the transition from a phobic to a philic state happens rather fast (within 1 s), while the inverse process takes comparative longer time ranging from ca. 80 to 200s. To be noticed, the short switching time is a big advantageous of our system, since previous *ex situ* superwetting conversion normally take rather long time as hours or days (*Angew. Chem. Int. Ed*, 2004, 130, 361-364; *Nature*, 1997, 388, 431; *Angew. Chem. Int. Ed*, 2018, 130, 5842-5847).

To emphasize it, we revised the figure caption of Figure S7.

Comment 4: Authors added the sentence “The concept is applicable to other systems as Cu/Pb, Cu/Sb, Au/Sn, Au/ Pb and Au/Sb for both polar and nonpolar oils, representing a new class of switchable surfaces”. This is a relevant addition showing generality. The only added data to back up this claim is the table in Figure 2d₃ and it shows only qualitatively that transition does occur. Do these systems reach the same contact angle values? Manuscript would be significantly improved if actual data of these systems is shown, such as contact angle vs cycles.

Answer 4: Here, in our manuscript, we claimed that “The concept is applicable to other systems as Cu/Pb, Cu/Sb, Au/Sn, Au/ Pb and Au/Sb for both polar and nonpolar oils, representing a new class of switchable surfaces” based on the experimental data. Here, we add the data in supplementary information of figure S8 and S9 to show the *in situ* reversible superwetting transition on other binary composite surface of Cu/Sb, Cu/Pb, Au/Sn, Au/Pb and Au/Sb. During the electrochemical deposition process, the metal atoms (Sn, Sb, Pb) were deposited onto either Cu or Au electrode, leading to a transition from a underwater superoleophobic to superoleophilic state; while after removing the potential, the spread oil thin film on the electrode is capable of turn back to the original spherical shape *in situ*, indicating a transition from a underwater superphilic to a superphobicity state (Figure S8). We further show the representative SEM images for the binary composite surface of Cu/Pb, Cu/Sb, Au/Sn, Au/ Pb and Au/Sb in Figure S9. Similar with that of Cu/Sn system, here only very thin layer of Sb, Pb or Sn was deposited on the highly-textured Cu (Au) surface, by which surface energy changes drastically but the surface roughness remains.

Comment 5: Supplementary movie 3 could show time indication.

Answer 5: Thanks for the kind suggestion, and we have revised movie 2 and movie 3 to show more clear the experimental process such as time.

REVIEWERS' COMMENTS:

Reviewer #1 (Remarks to the Author): The manuscript has improved greatly and is fit for publication

Answer #1: Thanks so much for the reviewer's positive evaluation on our manuscript.